# Assessment of Socioeconomic Dynamics and Electrification Progress in Tanzania Using VIIRS Nighttime Light Images

**Changjun Zhu** [1] , **Xi Li** [2,*] **and Yuanxi Ru** [1]

1   School of Remote Sensing and Information Engineering, Wuhan University, Wuhan 430079, China
2   State Key Laboratory of Information Engineering in Surveying, Mapping and Remote Sensing, Wuhan University, Wuhan 430079, China
*   Correspondence: lixi@whu.edu.cn; Tel.: +86-180-6241-1350

**Abstract:** Tanzania is one of the fastest-growing countries in the world, but it still faces many challenges of unbalanced development. However, Tanzania's economic assessment studies based on traditional statistics are mostly conducted at the national level, which leaves the details of regional economic disparity and electrification unknown. Despite experiencing one of the fastest urbanizations in the world, there is a lack of research on the match between urbanization and electrification in Tanzania. This study accesses the socioeconomic dynamics in Tanzania using nighttime light images from the Visible Infrared Imaging Radiometer Suite (VIIRS), providing spatiotemporal details for Tanzania's development. We examined the ability of nighttime light data to evaluate the socioeconomic dynamics in Tanzania and studied regional economic disparity based on the total nighttime light (TNL). Furthermore, the land electrification rate (LER) was defined to study the relationship between urbanization and electrification in Tanzania's major cities. We found that the LER was less than 0.9 in 2019 and had decreased from 2015 to 2019 in most cities, indicating that the power infrastructure gaps were widespread and growing in major cities. Additionally, we found a negative correlation between the change rate of land electrification and the urban expansion rate, indicating that the construction of power infrastructure has lagged behind the urbanization. We concluded that nighttime light data can effectively provide spatiotemporal details for socioeconomic dynamics in Tanzania. Additionally, our data mining method may be applied to other data-poor countries.

**Keywords:** VIIRS; nighttime light; Tanzania; black marble

## 1. Introduction

With the development of remote sensing, nighttime light remote sensing provides a new approach for socioeconomic research. The intensity of nighttime light radiance can reflect human socioeconomic activities and has been widely used in studies such as regional development [1,2], electricity estimation [3], and urbanization monitoring [4]. However, the majority of these studies are conducted in strong nighttime light radiation intensity areas [5], whereas only a small part of them focus on regions with a weak nighttime light radiation intensity [6–8]. There is a lack of socioeconomic statistics in these low electrification areas. However, according to a research conducted in Burkina Faso [9], most settlements in areas with low electrification rates and a low population density are undetectable to nighttime light sensors. As a result, it is necessary to exercise caution when employing nighttime light remote sensing as a socioeconomic proxy in these areas. Tanzania, a low electrification and data-poor country [10,11], is also in urgent need of socioeconomic assessment to support its sustainable development. However, few studies based on nighttime light remote sensing address Tanzania specifically [12].

Tanzania is a lower-middle income country in East Africa [13], consisting of Tanganyika and Zanzibar. With a stable political situation and abundant natural resources [10],

Tanzania is one of the fastest-growing countries in the world. However, there is significant regional economic disparity in Tanzania. First of all, Dar es Salaam, Tanzania's economic and industrial center, is growing much faster than the rest of the country [14]. In addition, because of the colonial history as well as the long-term agricultural and trade patterns, there is a significant north–south economic disparity in mainland Tanzania. Finally, the railway system has long been responsible for the majority of Tanzania's transportation, and has a considerable aggregation influence on the economy [15]. However, Tanzania's economic assessment studies based on traditional statistics are mostly conducted at the national level [16,17], which cannot reflect the regional economic disparity within the country.

Furthermore, electrification research in Tanzania faces similar problems. On the one hand, the government of Tanzania claims that economic growth is hampered by the lack of power infrastructure [18]. On the other hand, there is a lack of sufficient electrification statistics to guide policy making and investment in the power sector [12,19]. In Tanzania, nighttime light data have been used for electrification assessment [12]. However, the nighttime light data were collected by the Defense Meteorological Satellite Program's Operational Linescan System (DMSP/OLS), whose low light imaging capability is relatively poor [20]. Besides, DMSP/OLS data is only updated to 2013 and cannot reflect the latest electrification dynamics in Tanzania. In addition, since Tanzania has experienced rapid urbanization over the past decade, assessing the relationship between urbanization and electrification is critical for the country's sustainable development [21]. However, because of a lack of socioeconomic statistics and inefficient land-use classification, relevant research in Tanzania is restricted to a single city or region [22].

With a sufficient spatial and temporal resolution, nighttime light remote sensing has been extensively applied to socioeconomic assessment at the subnational level [1,23]. Additionally, by fusing with multi-temporal population and land data, nighttime light data have been used to monitor whether infrastructure is keeping pace with demographic and land transitions [24]. Thus, this paper aims to assess Tanzania's socioeconomic dynamics using nighttime light data collected by the Visible Infrared Imaging Radiometer Suite (VIIRS) from 2012 to 2020, providing the latest spatiotemporal details for Tanzania's sustainable development. First, based on the total nighttime light (TNL), the trend of regional economic disparity is analyzed. In addition, combining built-up area data and nighttime light data, the gap between electrification and urbanization is also examined. Our research will be beneficial to the formulation of Tanzania's Five-Year Development Plan (FYDP) and the assessment of its progress in achieving the United Nations Sustainable Development Goals (SDGs) [21,25].

## 2. Study Area and Data

### 2.1. Geography and Economy of Tanzania

Tanzania is located in Eastern Africa, south of the equator. It is bounded on the north by Kenya and Uganda, on the south by Zambia, Malawi, and Mozambique, on the west by Rwanda, Burundi, and the Democratic Republic of Congo, and on the east by the Indian Ocean. Tanzania is abundant in natural resources such as diamonds, gold, phosphates, and other minerals. Tanzania's political situation has long been stable, and its economy has grown substantially in recent years. It has been categorized as a lower-middle income country in 2020 [13]. According to the 2012 Tanzania National Census, Tanzania is divided into 30 regions (provinces), 25 of which are on the mainland Tanzania and 5 of which are in Zanzibar (the total number of regions increased to 31 in 2016 after the Songwe region was separated from the Mbeya region). Figure 1 shows the administrative division of Tanzania and its neighbors.

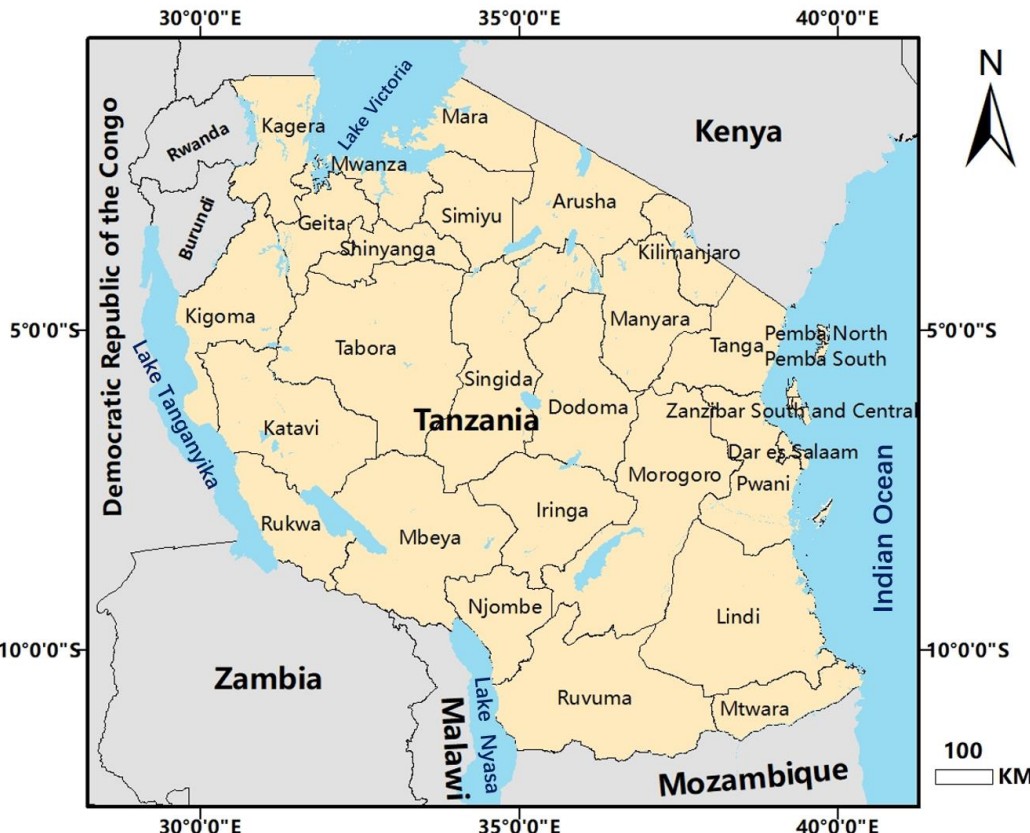

**Figure 1.** The administrative division of Tanzania and its neighbors.

### 2.2. Nighttime Light Data

In this study, we employed nighttime light data collected by the Day/Night Band (DNB) sensors of the Visible Infrared Imaging Radiometer Suite (VIIRS), on board the Suomi-National Polar-orbiting Partnership (S-NPP) satellite platform. Compared to the preceding DMSP/OLS nighttime light sensor, the VIIRS/DNB sensor is significantly superior in terms of a higher spatial resolution of 15 arc seconds, on-board calibration, increased sensitivity to low-level radiance, a larger radiation spectrum, etc. [26]. The latest VIIRS/DNB dataset is the Black Marble product suite provided by NASA. This product provides cloud-free, atmospheric-, terrain-, vegetation-, snow-, lunar-, and stray-light-corrected radiances, which can more accurately reflect human activities [27]. The all-angle snow-free layer in the Black Marble's annual moonlight-adjusted nighttime light product (VNP46A4) from 2012 to 2020 was used in this study.

### 2.3. Built-Up Area Data

In our study, the World Settlement Footprint (WSF) suite provided by the German Aerospace Center (DLR) was used as the built-up area data. The product, which is available as a global built-up area binary mask with a spatial resolution of 10 m, was generated by an advanced classification system using open-and-free optical and radar satellite imagery. Extensive validations have demonstrated that the WSF outperforms all other existing similar layers [28]. Currently, the WSF provided two products in 2015 and 2019.

### 2.4. Other Data

This study utilized a variety of auxiliary data, including administrative division data from the Database of Global Administrative Areas (GADM) [29], transportation network data from the Geofabric database [30], the urban boundary polygons and population data from Africapolis [31], precipitation data from the Climate Hazards Group InfraRed Precipitation with Station data (CHIRPS) global rainfall dataset [32], digital elevation model

data (DEM) from the Shuttle Radar Topography Mission (SRTM) [33], and GDP data from the National Bureau of Statistics of Tanzania [34], etc.

## 3. Methods

### 3.1. Regional Economic Disparity

3.1.1. Examining the Correlation between Nighttime Light and Economy

The total nighttime light (TNL) was defined as the sum of the digital number (DN) values within an administrative unit [35], namely:

$$\text{TNL} = \sum_{i=1}^{N} DN_i \tag{1}$$

where $DN_i$ represents the DN value of the pixel $i$ within an administrative unit. The DN value on the Black Marble image represents the radiance in units of $\text{nW} \cdot \text{cm}^{-2} \cdot \text{sr}^{-1}$. Numerous studies have demonstrated that the TNL can be used as an effective proxy for GDP in many regions [5], but its ability to be used as a socioeconomic proxy in less electrified areas has yet to be further tested [9]. Therefore, we performed a panel regression analysis of Tanzania's regional GDP and TNL, using the GDP data of 23 regions in mainland Tanzania from 2013 to 2019 provided by the National Bureau of Statistics of Tanzania. Considering the individual economic disparity between regions, we used a fixed effect variable intercept panel model:

$$GDP_{it} = \alpha_i + \beta TNL_{it} + \mu_{it} \tag{2}$$

where $GDP_{it}$ and $TNL_{it}$ represent the GDP and TNL of the region $i$ in the year $t$, respectively, $\alpha_i$ is the intercept of the fixed effect of the region $i$, and $\mu_{it}$ is the error term. Regression and correlation analyses were performed on this model, and then the regional economic disparity was investigated.

3.1.2. Regional Economic Disparity Analysis Based on the TNL

As the results of Section 4.1.1 showed that there is a strong positive correlation ($R^2 = 0.89$) between the regional (provincial) TNL and GDP, this section investigated regional economic spatiotemporal disparity based on the TNL.

First of all, to study the economic dynamics of Dar es Salaam, the city primacy index was calculated by dividing the Dar es Salaam's TNL by the national TNL, namely:

$$R_{dar} = \frac{TNL_{dar}}{TNL_{nat}} \tag{3}$$

where $R_{dar}$ represents the city primacy index of Dar es Salaam, $TNL_{dar}$ is the TNL of Dar es Salaam, and $TNL_{nat}$ is the national TNL.

In addition, to investigate the north–south economic disparity, we first divided all regions into southern and northern regions based on the relative north–south position of the regional capitals on the central railway. Because Dar es Salaam has an excessively high GDP [34], including it on either side of the north and south will result in an imbalance in the north–south economic ratio; hence, Dar es Salaam was not included on either side. Then, the ratio of the total TNL of the southern regions to the TNL of mainland Tanzania excluding Dar es Salaam, denoted as $R_{sou}$, was calculated by the following equation:

$$R_{sou} = \frac{TNL_{sou}}{TNL_{mai} - TNL_{dar}} \tag{4}$$

where $TNL_{sou}$, $TNL_{mai}$, and $TNL_{dar}$ are the TNL of the southern regions, mainland Tanzania, and Dar es Salaam, respectively.

Finally, to study the economic aggregation effect of the railway system, we established a railway buffer with a distance of 20 km, which would cover approximately all the cities

connected to the railway. Then, the ratio of the TNL in the railway buffer excluding Dar es Salaam to the TNL in mainland Tanzania except for Dar es Salaam, denoted as $R_{rai}$, was calculated by the following equation:

$$R_{rai} = \frac{TNL_{rai} - TNL_{dar}}{TNL_{mai} - TNL_{dar}} \tag{5}$$

where $TNL_{rai}$, $TNL_{dar}$, and $TNL_{mai}$ are the TNL of the railway buffer, Dar es Salaam, and mainland Tanzania, respectively.

### 3.2. Power Infrastructure in Major Cities

3.2.1. Land Electrification Rate

To investigate the spatial expansion of the urban power infrastructure, we defined the land electrification rate (LER) as the proportion of the urban built-up area with power infrastructure in the total urban built-up area (Figure 2), namely:

$$LER = \frac{A_p}{A_t} \tag{6}$$

where $A_t$ is the area of the total urban built-up area and $A_p$ is the area of the built-up area with power infrastructure.

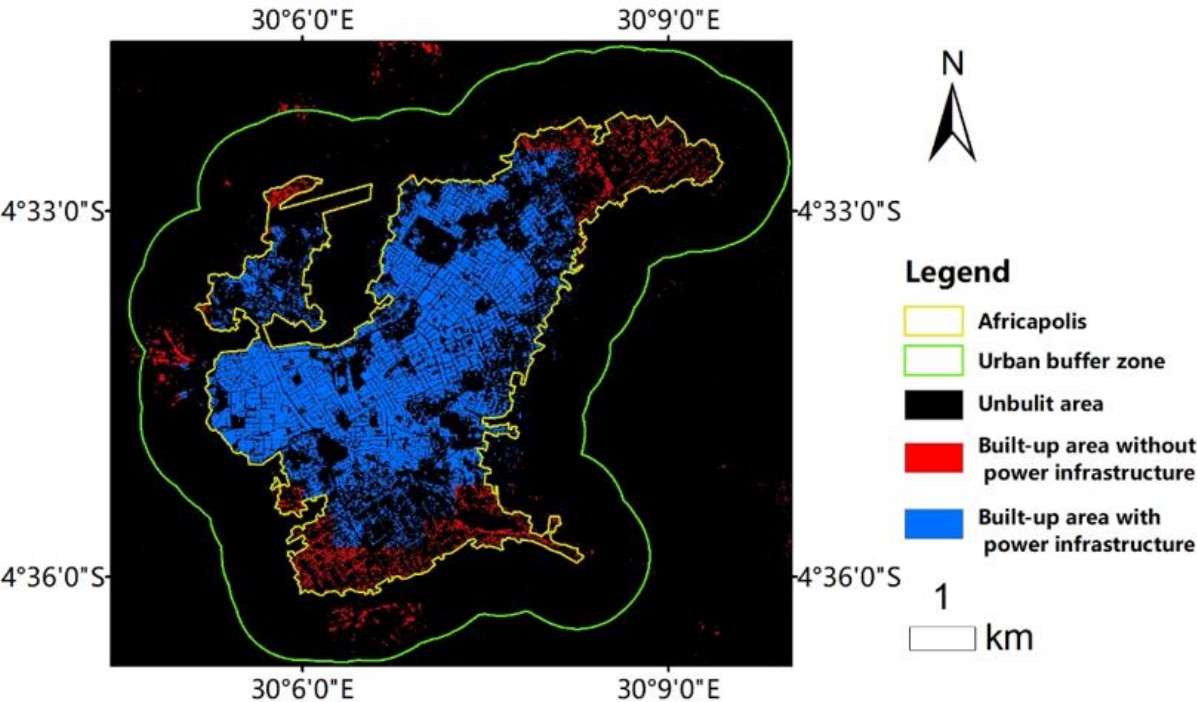

**Figure 2.** Illustration for the land electrification rate (LER). LER was defined as the proportion of the urban built-up area with power infrastructure (blue area) in the total urban built-up area (blue and red areas).

The urban built-up area was extracted using the WSF layer. Since a specific threshold was set in the annual Black Marble products to remove background noise [27], we assumed that within the urban built-up area, the area with positive radiance is the area with power infrastructure. The city boundaries were determined using the Africapolis urban polygons. Since the polygons may underestimate the actual city boundaries, we created buffers based on the polygons. The buffer distances were determined based on the city population level. To avoid subjectivity in setting the buffer distances, we used two groups of buffer distances as controls (Table 1). The distances were all set to encompass the city boundaries and to not

intersect other cities. Additionally, the distances of the second group were slightly larger than that of the first group, so as to compare the electrification between the city edge and the city center. According to the 2015 urban population data provided by Africapolis, a total of 20 cities in Tanzania with a population more than 100,000 were selected. Figure 3 shows the distribution of the selected major cities in Tanzania.

**Table 1.** Buffer distances were set according to the city population level. Two groups of buffer distances were set, buffer distances 1 and buffer distances 2.

| Population (million) | Number of Cities | Buffer Distances 1 (km) | Buffer Distances 2 (km) |
|---|---|---|---|
| 0.1~0.2 | 10 | 1 | 3 |
| 0.2~0.5 | 4 | 2 | 4 |
| 0.5~1.0 | 4 | 3 | 5 |
| >1.0 | 1 | 6 | 8 |

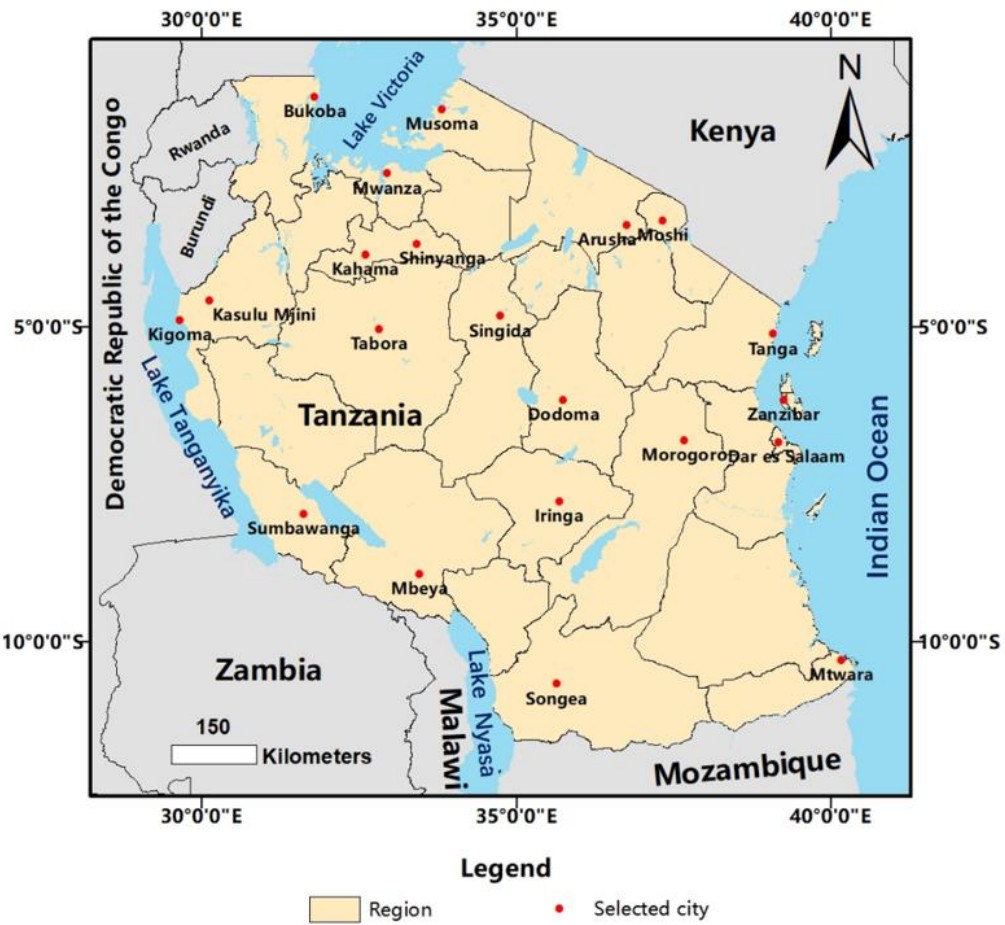

**Figure 3.** The distribution of selected major cities in this study. The selected cities all have populations of more than 100,000.

### 3.2.2. Spatiotemporal Dynamics of Land Electrification

Since the WSF layer is only available in 2015 and 2019, we calculated the land electrification rate (LER) for each city in these two years. To investigate the progress of urban power infrastructure construction, the change rate of land electrification, denoted as $\Delta L$, was defined as:

$$\Delta L = \frac{LER_{2019} - LER_{2015}}{LER_{2015}} \tag{7}$$

where $LER_{2015}$ and $LER_{2019}$ represent the LER in 2015 and 2019, respectively.

To explore the factors influencing power infrastructure construction, we analyzed the correlations between the change rate of land electrification and the following variables: the urban expansion rate, the change rate of the TNL, and the steepness of the urban terrain.

The urban expansion rate, used to measure the change rate of the built-up area, was calculated by the following equation:

$$\Delta A_t = \frac{A_{t,2019} - A_{t,2015}}{A_{t,2015}} \tag{8}$$

where $A_{t,2015}$ and $A_{t,2019}$ represent the areas of the total urban built-up area in 2015 and 2019, respectively, and $\Delta A_t$ is the urban expansion rate.

The change rate of the TNL, denoted as $\Delta TNL$, was calculated by the following equation:

$$\Delta TNL = \frac{TNL_{2019} - TNL_{2015}}{TNL_{2015}} \tag{9}$$

where $TNL_{2015}$ and $TNL_{2019}$ are the TNL of the city in 2015 and 2019, respectively.

The steepness of the urban terrain was defined as the standard deviation of the elevation in the urban buffer, namely:

$$S = \sqrt{\frac{1}{N}\sum_{j=1}^{N}\left(h_j - \bar{h}\right)^2} \tag{10}$$

where $S$ represents the steepness of the urban terrain and $\bar{h}$ is the average elevation of the city.

## 4. Results

### 4.1. Results of Regional Economic Disparity

#### 4.1.1. Results of Correlation Analysis between TNL and GDP

Table 2 shows the regression results of Tanzania's regional TNL and GDP based on a fixed effects variable intercept panel model. The regression analysis showed that there is a strong positive correlation between the GDP and the TNL, with an $R^2$ of 0.894 ($p < 0.01$). Despite Tanzania's low rate of electrification and predominance of agriculture [11,34], nighttime light data are still accurate predictors to estimate socioeconomic parameters for the country. As a result, the TNL can be used to monitor the spatial and temporal disparity of the regional economies in Tanzania.

**Table 2.** Panel regression results of Tanzania regional total nighttime light (TNL) and GDP based on fixed effects variable intercept panel model.

| Variable | Coefficient |
|---|---|
| Constant term | 1107.162 *** |
|  | (134.107) |
| $\beta$ | 0.052 *** |
|  | (0.001) |
| Time effects | No |
| Region effects | Yes |
| Observations | 184 |
| Regions | 23 |
| $R^2$ | 0.894 |

Notes: Robust standard errors in brackets, *** $p < 0.01$.

#### 4.1.2. Results of Regional Economic Disparity Analysis Based on TNL

The administrative boundary of Dar es Salaam is shown in Figure 4. In addition, Figure 5 shows the city primacy index of Dar es Salaam from 2012 to 2020, namely the proportion of the Dar es Salaam's TNL in the national TNL during the period. The proportion

was around 45% in 2012, despite Dar es Salaam accounting for only 10% of the national population at the time [34], reflecting Dar es Salaam's high urban priority and Tanzania's significant regional population and economic imbalance. Moreover, the proportion was still rising from 2012 to 2020, indicating that the city's economy is growing faster than the rest of the country. In addition, the city primacy index rose with fluctuation and decreased in 2015 and 2018, respectively. Additionally, the drops coincided with the years of severe flooding, indicating that the economic growth could be jeopardized by floods [36]. To promote sustainable urban economic growth, the governments should improve urban resilience, especially in vulnerable informal settlements.

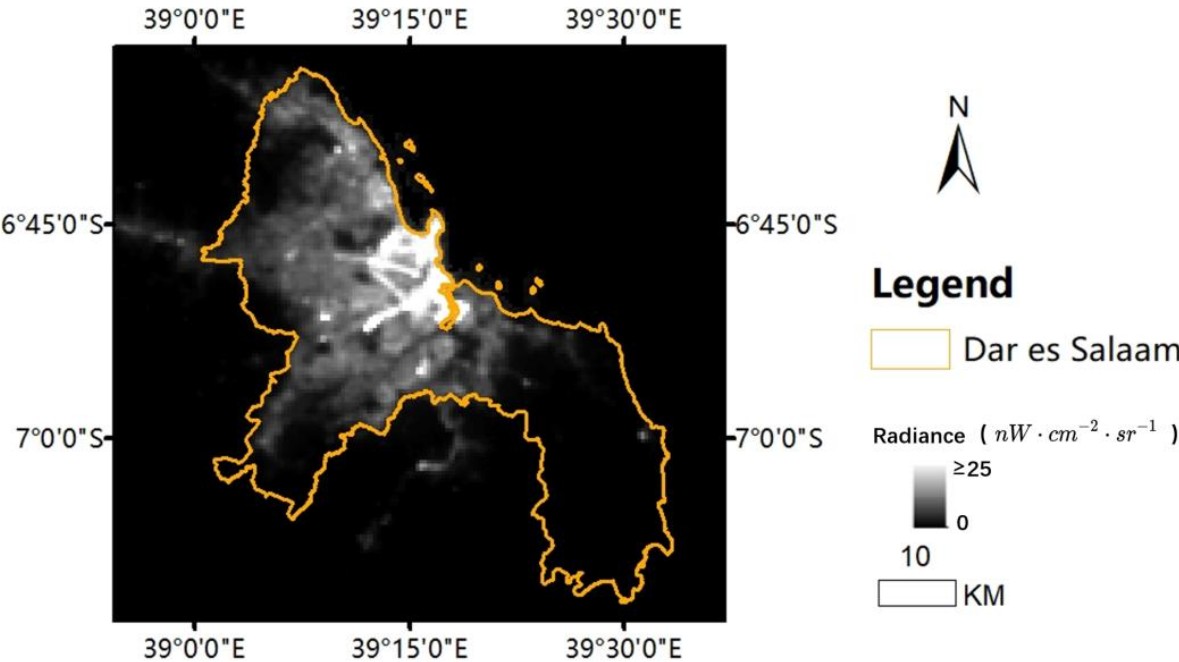

**Figure 4.** The administrative boundary of Dar es Salaam (yellow polygon) and Black Marble data in 2020 (grey-scale background).

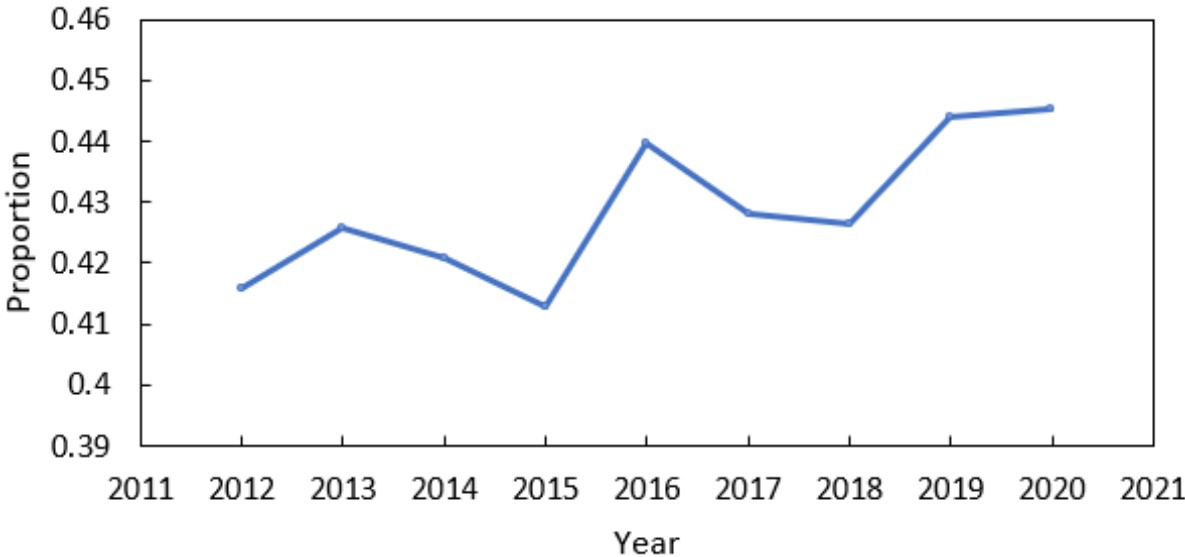

**Figure 5.** The proportion of the total nighttime light (TNL) of Dar es Salaam in the nation's TNL from 2012 to 2020.

The division result of the northern and southern regions of mainland Tanzania is shown in Figure 6, and the proportion of the total TNL of the southern regions to the TNL of mainland Tanzania excluding Dar es Salaam from 2012 to 2020 is shown in Figure 7. This proportion is around 30%, indicating that the economic size of the south is significantly smaller than that of the north. However, the proportion shows a slight upward trend, indicating that the economic growth rate of the south was slightly faster than that of the north, and that the economic gap between the north and the south was gradually narrowing. This may have benefited from a series of pro-southern policies in recent years, such as upgrading the port of Mtwara [37], a key southern port. In addition, active offshore LPG and gas exploration activities in the south and a significant increase in graphite production in the Lindi region are also likely to contribute to the southern economic boom [38].

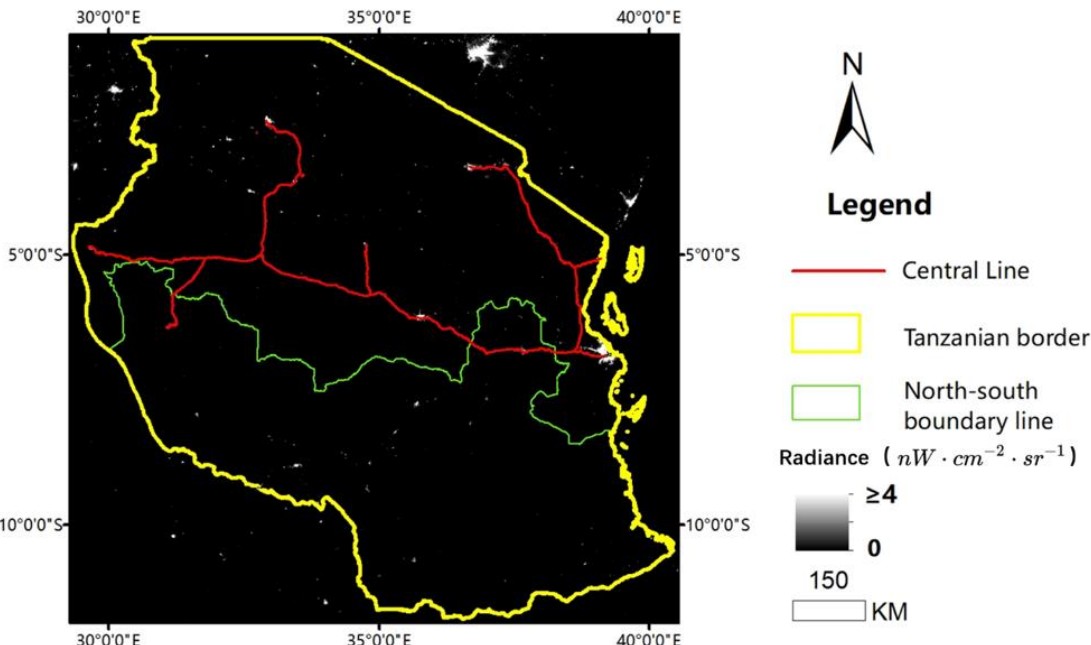

**Figure 6.** The division results of northern and southern regions of mainland Tanzania, 2020 Black Marble data in the background.

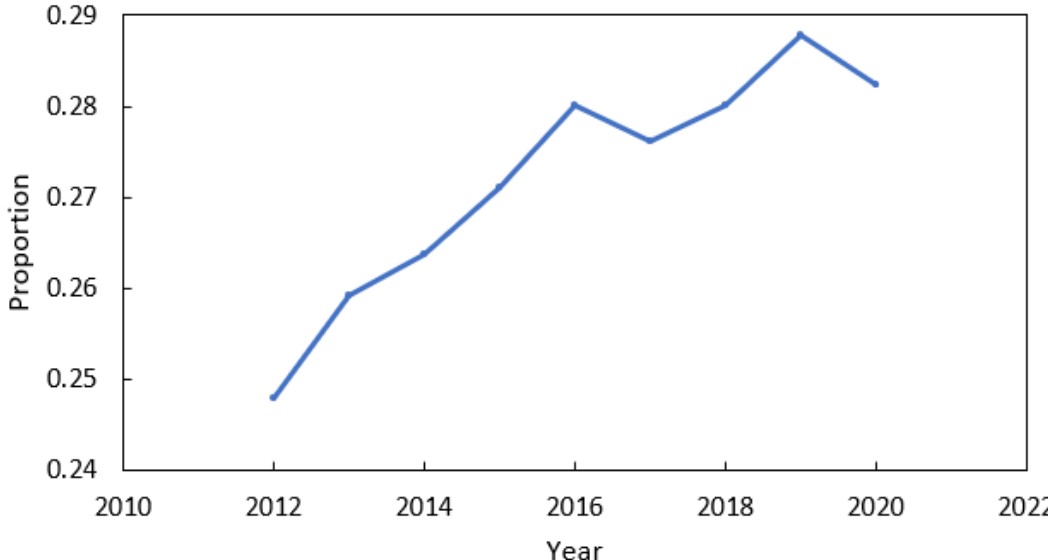

**Figure 7.** The proportion of total nighttime light (TNL) of southern regions to TNL of mainland Tanzania excluding Dar es Salaam from 2012 to 2020.

Figure 8 shows a 20 km railway buffer. We then calculated the ratio of the TNL of the railway buffer excluding Dar es Salaam to the TNL of mainland Tanzania excluding Dar es Salaam from 2012 to 2020 (Figure 9). The ratio fluctuated slightly over time, showing a downward trend from 2012 to 2018 and an upward trend from 2018 to 2020. This trend may have been caused by the following factors: (1) the disrepair of the railway and the deteriorating operation conditions may have contributed to the downward trend [39]; (2) due to the diverting effect of the government's vigorous construction of inter-regional highways, the aggregation effect of the railway on the economy gradually declined from 2012 to 2018 [40]; and (3) the Standard Gauge Railway (SGR) program might have played a role in the rise trend. The SGR aims to upgrade the country's original railways and replace the old and inefficient meter-gauge railway system with the international Standard Gauge Railway [41].

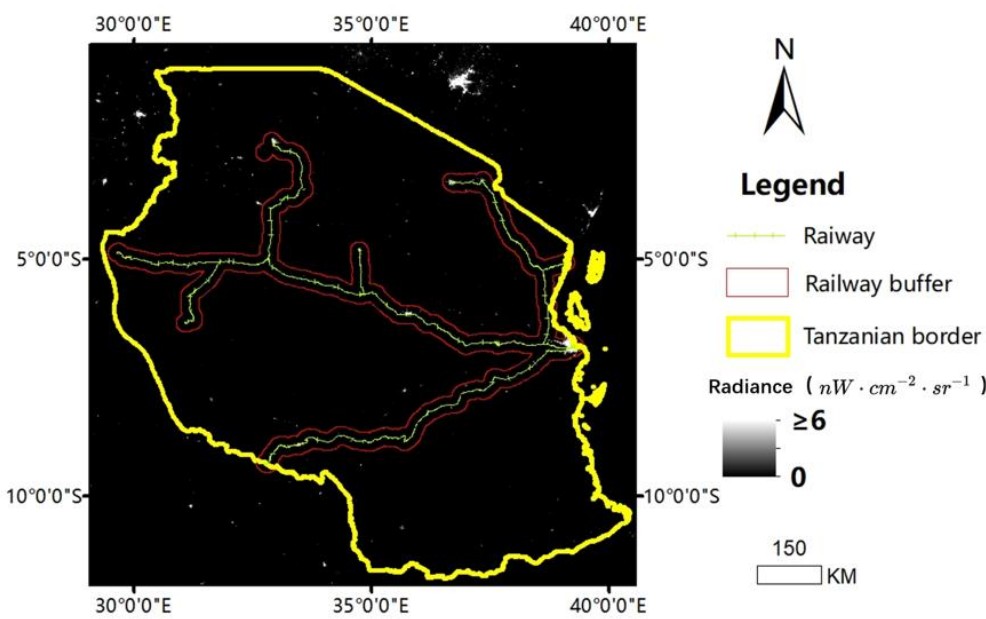

**Figure 8.** A 20 km railway buffer (red polygon) and Black Marble data in 2020 (grey-scale background).

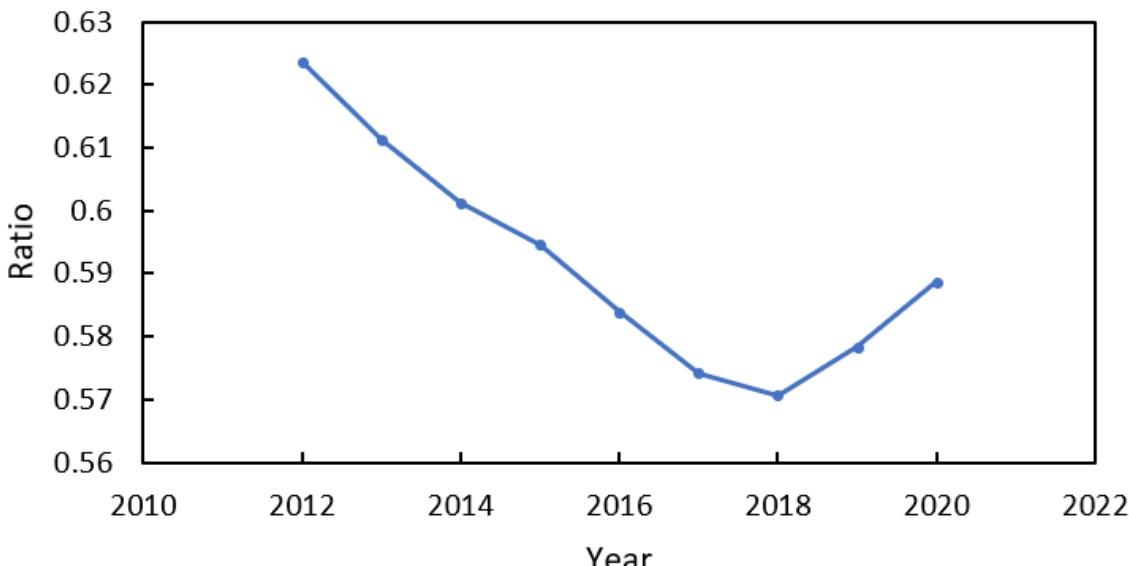

**Figure 9.** The ratio of the total nighttime light (TNL) of the railway buffer excluding Dar es Salaam to the TNL of mainland Tanzania excluding Dar es Salaam from 2012 to 2020.

*4.2. Temporal and Spatial Dynamics of Power Infrastructure in Major Cities*

4.2.1. Results of Land Electrification Rate

Figure 10 shows two groups of buffers established for some cities, with buffer 1 being smaller than buffer 2. Each city's land electrification rate (LER) was calculated in two groups of buffers in 2015 and 2019. We then analyzed the spatial and temporal characteristics of the LER, respectively.

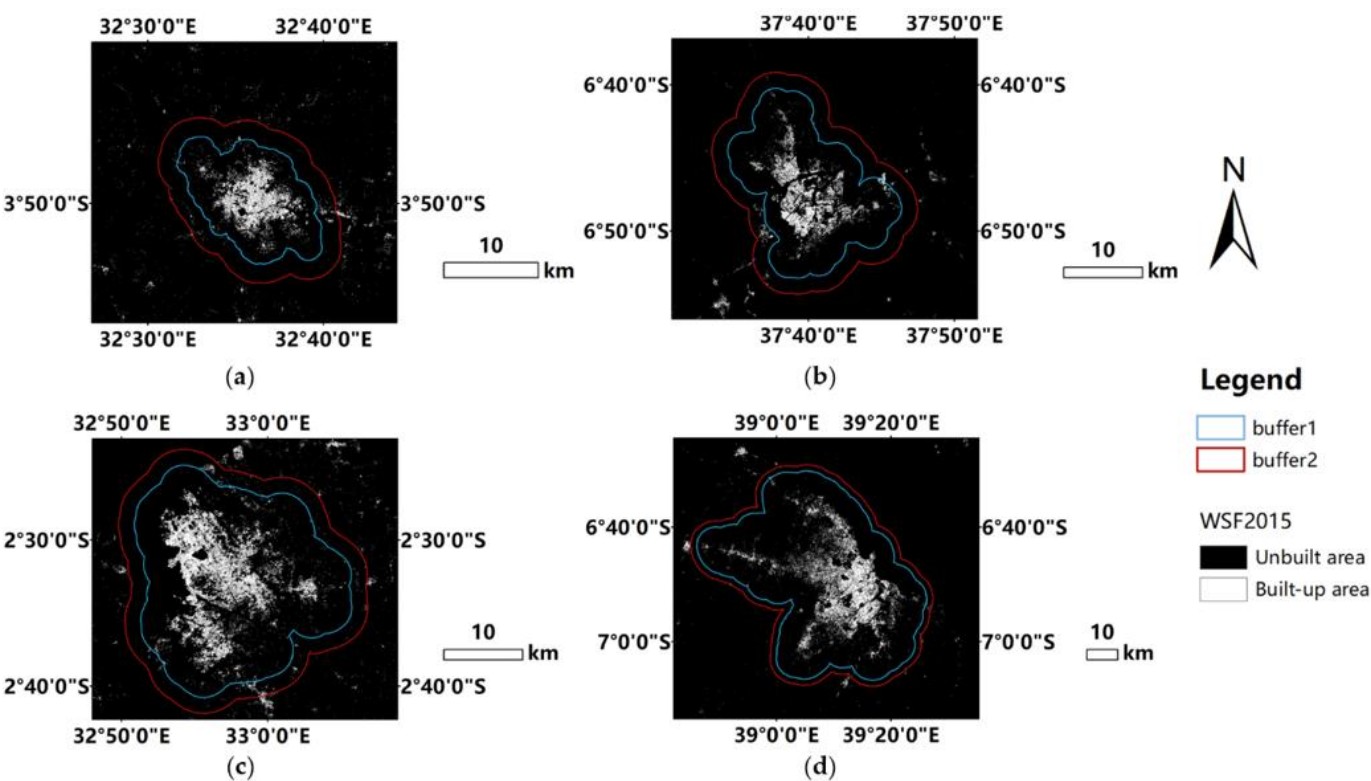

**Figure 10.** Buffers for some cities: (**a**) Kahama; (**b**) Morogoro; (**c**) Mwanza; (**d**) Dar es Salaam. Two buffers of different sizes were created for each city. The binary background is the 2015 World Settlement Footprint (WSF2015) layer.

Figures 11 and 12 demonstrate the comparison of the LER in different years within the same group of buffers. The findings from the above figures are as follows: (1) within the same year, most of the 20 cities in either group of buffers had an LER less than 95% in 2015 and less than 90% in 2019, which indicates that there are widespread gaps between urbanization and power infrastructure construction in Tanzania's major cities; (2) in all but four of the twenty cities, the LER in 2015 was higher than the LER in 2019, indicating that the gaps widened from 2015 to 2019.

Figures 13 and 14 show the comparison results of the LER in the same years within different group of buffers. The findings from the above figures are as follows: (1) the LER of buffer 2 was lower than the LER of buffer 1 in 2015 and 2019, while buffer 2 was larger than buffer 1, indicating that power infrastructure was poorer in the edge of the city than in the center of the city; (2) in most cities, the LER difference between the two groups of buffers in 2019 was greater than that in 2015, indicating that the power exacerbated between the suburbs and the central cities increased from 2015 to 2019.

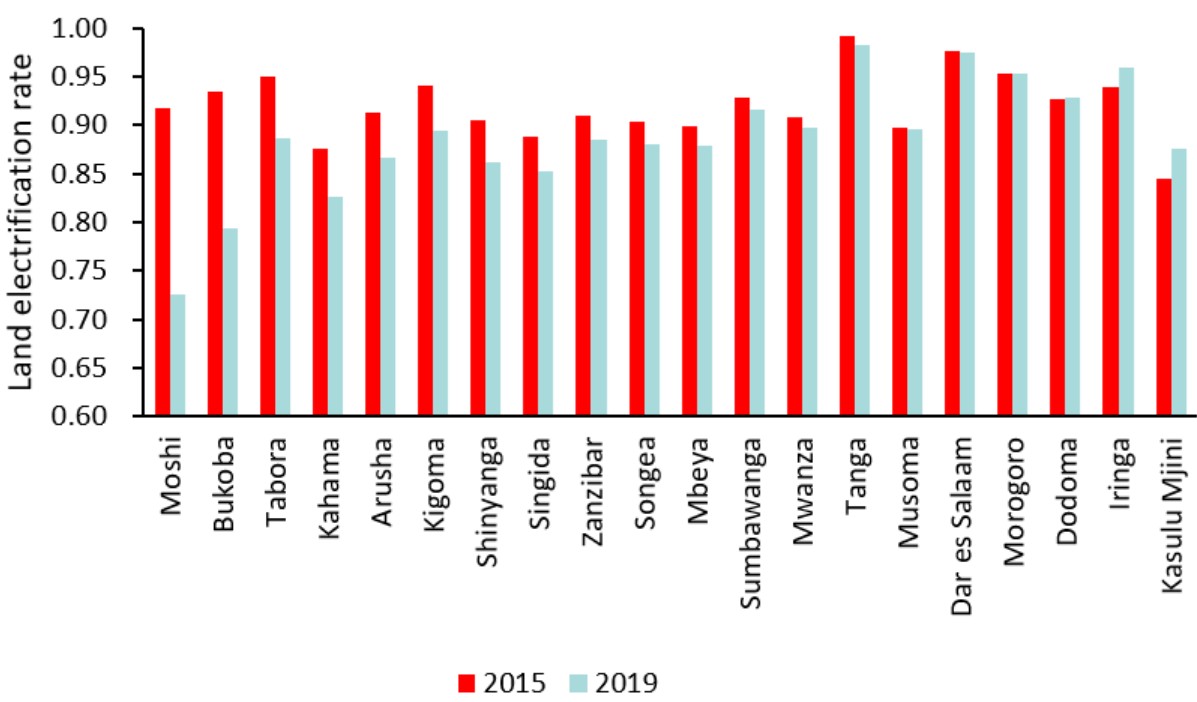

**Figure 11.** The first group of buffers' land electrification rates in 2015 and 2019.

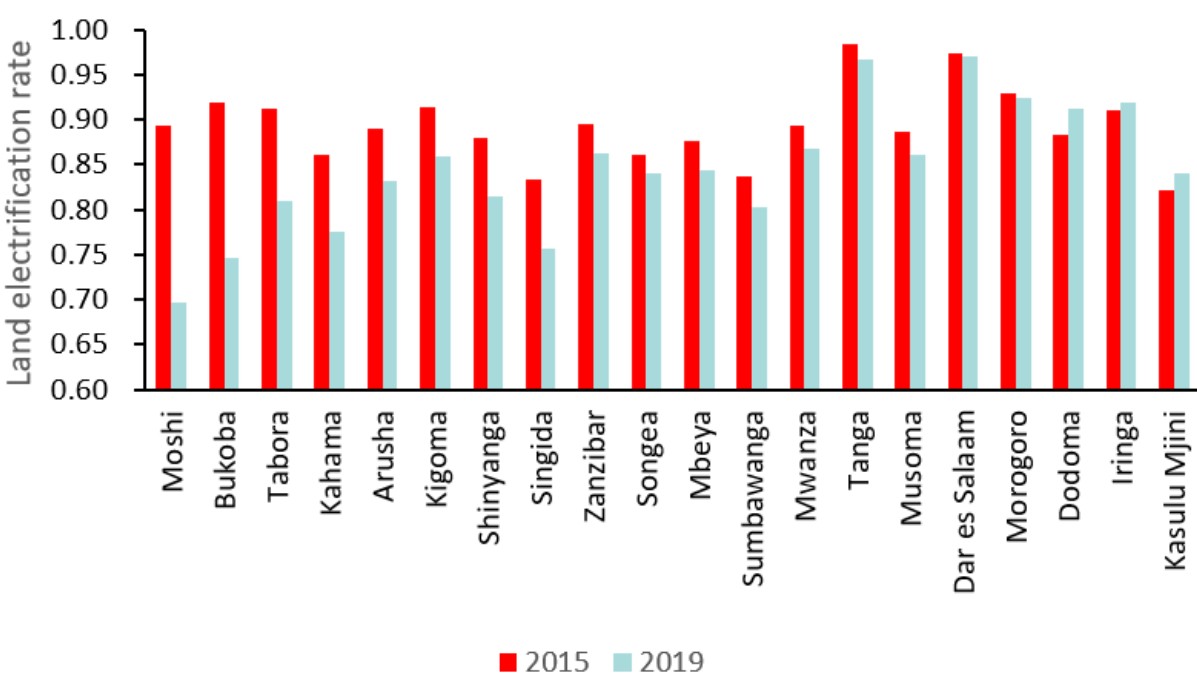

**Figure 12.** The second group of buffers' land electrification rates in 2015 and 2019.

4.2.2. Correlation Analysis between the Change Rate of Land Electrification and Other Factors

This section took the change rate of the land electrification of each city in buffer 1 as an example to study its correlations with other variables. The urban expansion rate, change rate of the TNL, and the steepness of the urban terrain of each city in buffer 1 were calculated, respectively, as shown in Table 3.

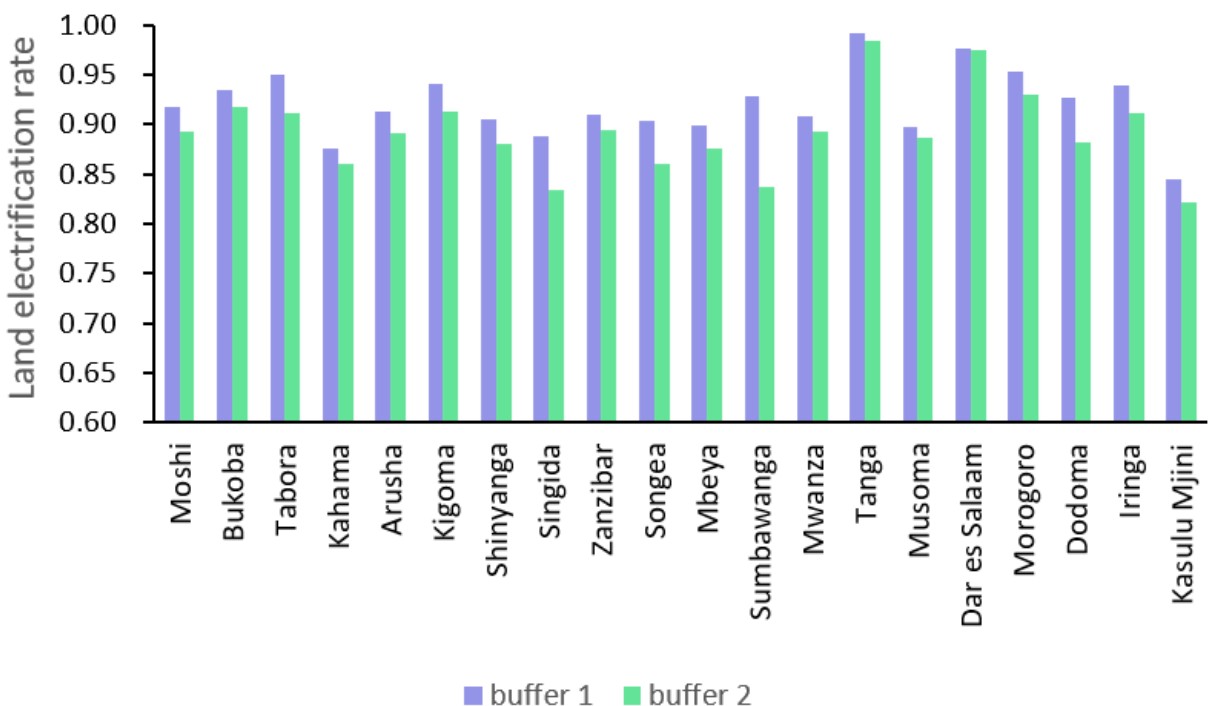

**Figure 13.** Two groups of buffers' (buffer 1 and buffer 2) land electrification rates in 2015.

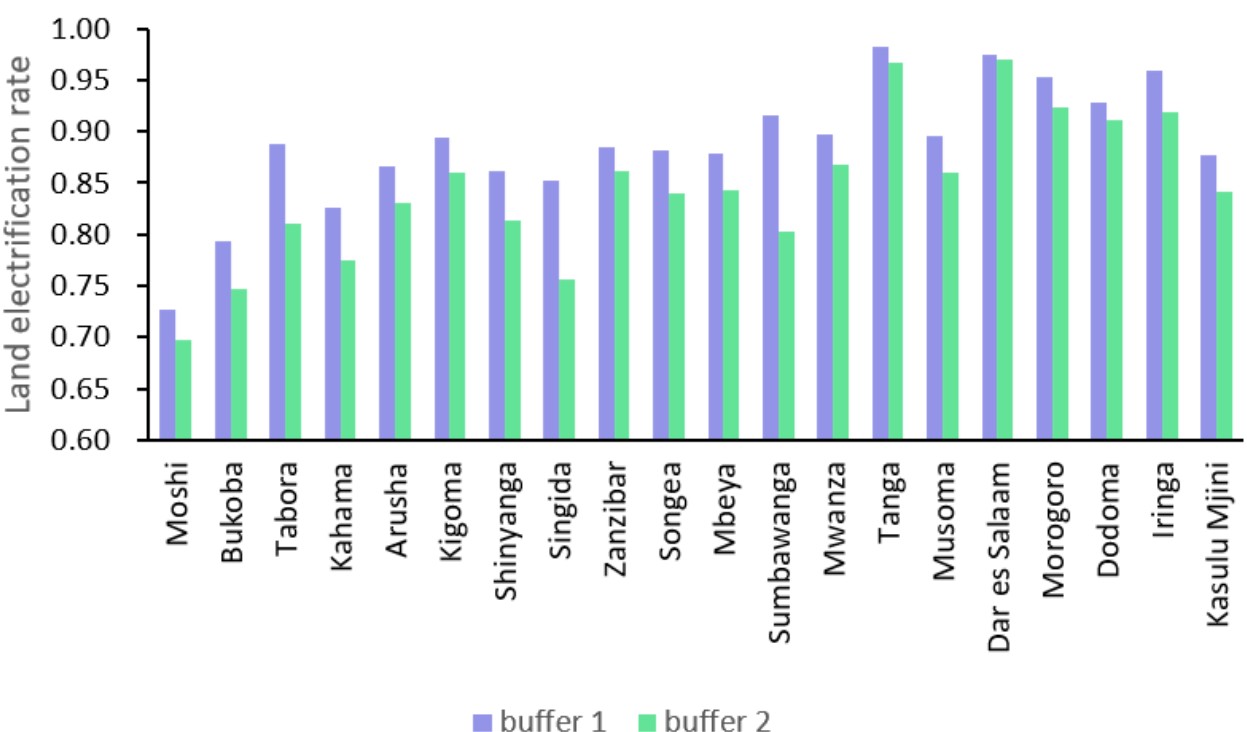

**Figure 14.** Two groups of buffers' (buffer 1 and buffer 2) land electrification rates in 2019.

**Table 3.** The change rate of land electrification, the urban expansion rate, the change rate of TNL, and the steepness of the urban terrain of each city in buffer 1.

| City | Change Rate of Land Electrification | Urban Expansion Rate | Change Rate of TNL | Steepness of the Urban Terrain (m) |
|---|---|---|---|---|
| Moshi | −0.209 | 0.50 | −0.20 | 347.15 |
| Bukoba | −0.151 | 0.85 | 0.09 | 62.98 |
| Tabora | −0.065 | 0.42 | 0.17 | 20.25 |
| Kahama | −0.057 | 0.64 | 0.34 | 23.17 |
| Arusha | −0.052 | 0.32 | 0.15 | 273.75 |
| Kigoma | −0.049 | 0.21 | −0.16 | 72.39 |
| Shinyanga | −0.049 | 0.28 | 0.26 | 17.09 |
| Singida | −0.040 | 0.35 | 0.11 | 24.06 |
| Zanzibar | −0.028 | 0.30 | 0.38 | 22.53 |
| Songea | −0.026 | 0.31 | 0.32 | 56.54 |
| Mbeya | −0.023 | 0.26 | 0.38 | 227.05 |
| Sumbawanga | −0.014 | 0.33 | 0.39 | 48.53 |
| Mwanza | −0.012 | 0.39 | 0.37 | 48.93 |
| Tanga | −0.009 | 0.25 | 0.01 | 22.46 |
| Musoma | −0.002 | 0.35 | 0.25 | 43.86 |
| Dar es Salaam | −0.001 | 0.28 | 0.39 | 56.94 |
| Morogoro | 0.000 | 0.24 | 0.18 | 185.10 |
| Dodoma | 0.002 | 0.39 | 0.60 | 45.10 |
| Iringa | 0.021 | 0.17 | 0.44 | 67.34 |
| Kasulu Mjini | 0.038 | 0.19 | 0.65 | 55.11 |

The linear regression results of the urban expansion rate and the change rate of land electrification are shown in Figure 15. The regression analysis showed that urban expansion rate was negatively correlated with the change rate of land electrification, with an $R^2$ of 0.4816 ($p < 0.01$). Given that the land electrification rate was defined as the proportion of urban built-up area with power infrastructure to the total urban built-up area, the negative correlation indicates that the proportion of the city with power infrastructure decreased with the increase in urban expansion. In addition, the construction of power infrastructure has lagged behind the urbanization.

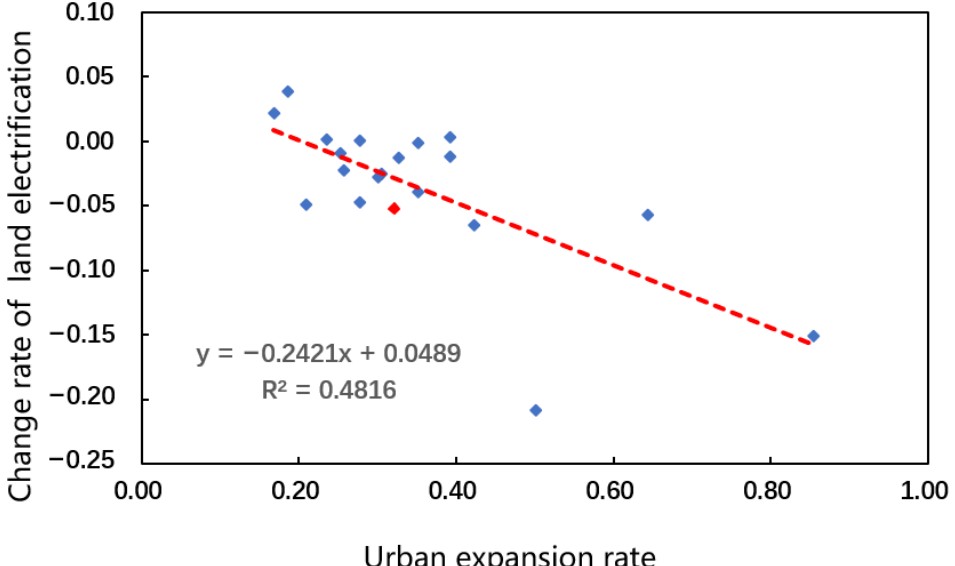

**Figure 15.** The scatter diagram showing the relationship between urban expansion rate and the change rate of land electrification. The outlier Moshi was marked red, which suffered from a significant decline in the total nighttime light.

Figure 16 shows the linear regression results of the change rate of the TNL and the change rate of land electrification. The regression analysis showed that the change rate of the TNL was positively correlated with the change rate of land electrification, with an $R^2$ of 0.4881 ($p < 0.01$). Since there is a strong positive correlation between the GDP and the TNL, the power infrastructure popularization is consistent with the city's economic development.

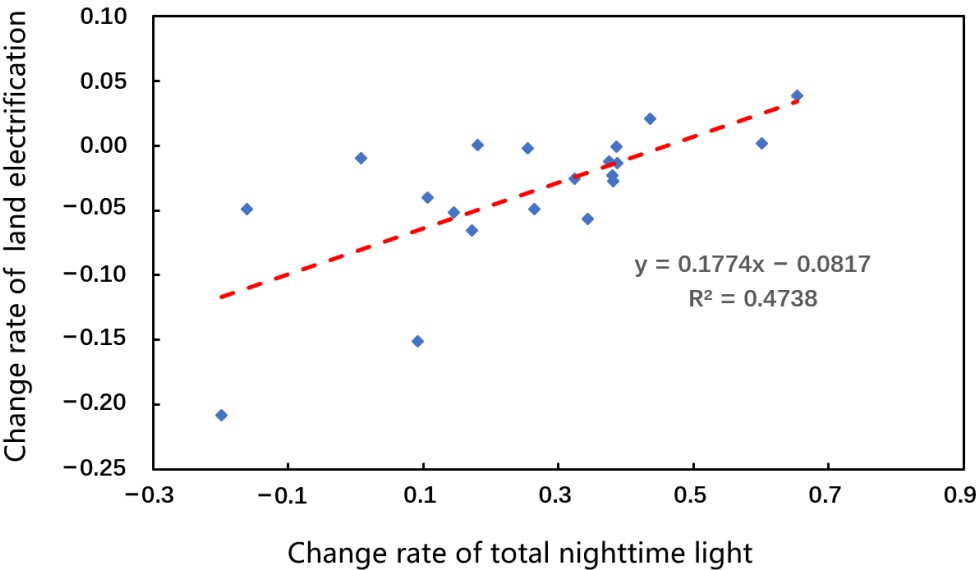

**Figure 16.** The scatter diagram showing the relationship between the change rate of total nighttime light and the change rate of land electrification.

Note that in Figure 16, Kigoma has a negative change rate of the TNL and a significant decrease in the LER. Kigoma is one of the busiest port cities on the northeast side of Lake Tanganyika. However, due to decreased water depth at the docks and declining fish populations [42], the economic growth of the city has been jeopardized, resulting in a decline in the city's TNL.

There is no significant linear relationship between the steepness of the urban terrain and the change rate of land electrification. However, with the increase in the steepness of the urban terrain, the change rate of land electrification has a general tendency to increase first and then decrease. Furthermore, Bukoba is an outlier in this trend, with a much lower change rate of land electrification than other cities with a similar steepness of urban terrain. This could have been caused by the September 2016 earthquake, which was the largest and deadliest in Tanzania since 2000. Bukoba was the most severely damaged city in the earthquake, with at least 840 houses destroyed and another 1264 severely damaged, leaving thousands homeless [43].

## 5. Discussion

### 5.1. Advantages of Nighttime Light Remote Sensing

Tanzania is one of the fastest-developing countries in the world [10], but there are still a number of unbalanced development problems [11,44,45]. Therefore, relevant socioeconomic research is urgently needed to support the country's sustainable development. On the one hand, traditional statistics in Tanzania are severely lacking. On the other hand, the study area based on remote sensing data is typically limited to certain cities [14,22,46–50]. The existing research is insufficient to capture Tanzania's entire socioeconomic dynamics. This study employed nighttime light remote sensing to assess the overall socio-economic dynamics of Tanzania with a high temporal and spatial resolution, effectively filling the gaps in relevant research.

### 5.2. The Relationship between Nighttime Light Data and Power Infrastructure

Our study investigated urban power infrastructure using nighttime light data. However, the relationship between power infrastructure and nighttime light data requires further clarification. As urban nighttime light mainly comes from buildings and street lamps, there is a direct correlation between nighttime light remote sensing and electric lighting infrastructure [7]. According to research conducted in rural Vietnam, the brightness of villages is an increasing function of both the number of electrified homes and the number of streetlights [51]. In our study, the land electrification rate can be regarded as the proportion of the city with access to electric lighting. Furthermore, because the electricity for buildings and streetlamps is generated by power stations, nighttime light remote sensing is implicitly correlated with power generation infrastructure. Therefore, the land electrification rate is also a useful proxy for the overall urban power generation capacity.

### 5.3. The Outlier Moshi

Note that there is an obvious outlier, Moshi, in Figure 15, whose urban expansion rate is not very high, but the change rate of land electrification is surprisingly low. Located at the foot of Mount Kilimanjaro, a popular tourist destination, Moshi receives nearly one million visitors a year and has one of the highest per capita incomes in Tanzania. However, Moshi's TNL in 2019 was nearly 20% lower than in 2015, showing an unusual decline. This may have been caused by the flood in the area [52,53]. Figure 17 shows that the year when Moshi's TNL decreased corresponds to the year when rainfall suddenly increased. Figure 18 shows that areas with steep terrain tend to suffer from nighttime light decline. Moshi's urban settlements are densely distributed on the hillside. It is likely that floods caused by heavy rains have destroyed a large number of settlements, as well as the power infrastructure, resulting in a significant decrease in the TNL and the LER.

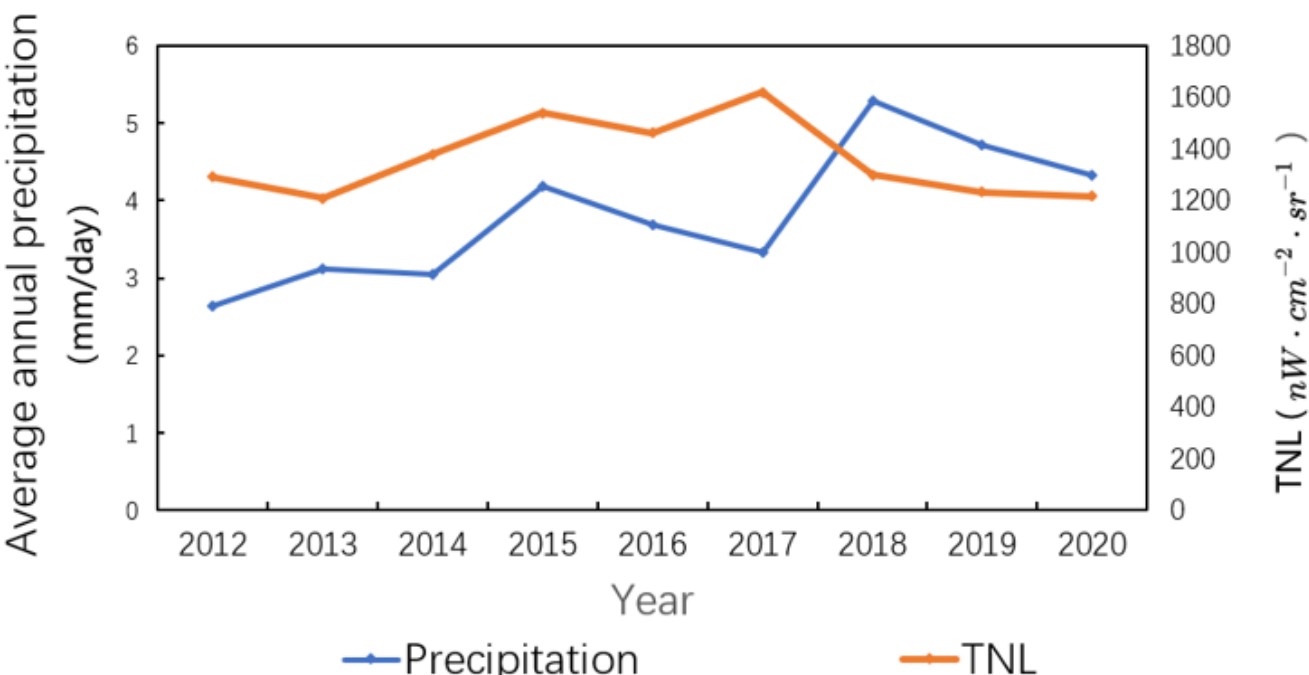

**Figure 17.** Total nighttime light (TNL) and average annual precipitation of Moshi.

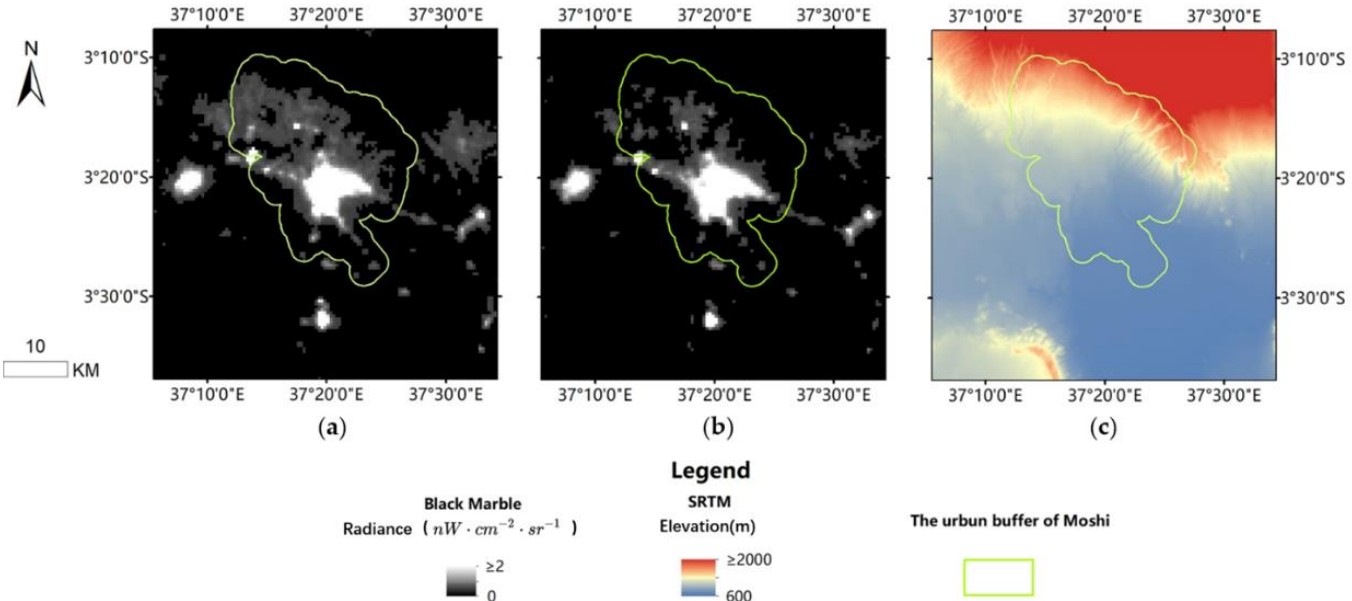

**Figure 18.** Comparison of the terrain and nighttime light decrease in Moshi. (**a**) Black Marble data in 2015; (**b**) Black Marble data in 2019; (**c**) SRTM DEM data.

### 5.4. Limitations and Future Work

However, some limitations should be noted. With a minimum nighttime light detection limit and a satellite overpass time of 1:30 midnight [27], the VIIIRS/DNB sensor is unable to detect the feeble and transient nighttime light emitted by human activity. Thus, nighttime light remote sensing has limited capabilities to monitor human activities in sparsely populated rural communities [9]. However, the positive correlation between GDP and TNL increases as research units are aggregated to higher administrative levels [23], where the TNL reflects the sum of various economic activities with different population densities. In our study, there is a strong correlation ($R^2 = 0.894$) between the TNL and GDP at the regional (provincial) level in Tanzania, allowing us to investigate regional economic disparity. Nevertheless, it is challenging to evaluate socioeconomic activities at the lower administrative levels in low-electrification areas simply using nighttime light data [7]. Therefore, in our study of city-level electrification, we used built-up area data, which effectively assisted nighttime light remote sensing for socioeconomic dynamic assessment. For example, Bukoba's economic recession caused by an earthquake was not obviously reflected in the TNL (but rather a TNL increase between 2015 and 2019). However, the significant mismatch between the TNL growth and the rapid expansion of the built-up areas reveals Bukoba's economic unrest. As a result, the use of auxiliary data will improve the socioeconomic assessment in low-electrification areas. Future work should apply more auxiliary data related to nighttime light, such as land surface temperature data and urban green space datasets [54,55], to better monitor socioeconomic dynamics in low-electrification areas.

In our study, the all-angle layer in the Black Marble annual composites were used to assess the socioeconomic dynamics of Tanzania. However, the composites have some limitations. First of all, the use of an all-angle layer ignores the angular effect on nighttime radiance [56,57], namely, satellite-observed artificial light radiance varies in relation to the satellite viewing angles. In addition, the time span (2012–present) of the VIIRS composites is relatively short. In future work, the use of intercalibrated nighttime light composites between the DMSP/OLS and VIIRS and the consideration of angular effects will facilitate the accurate and long-term socioeconomic assessment [58].

## 6. Conclusions

This paper carried out a socioeconomic assessment of Tanzania based on time series Black Marble nighttime light images. We found a strong correlation ($R^2 = 0.89$) between regional total nighttime light (TNL) and GDP in Tanzania, indicating that the TNL is still a good proxy for GDP in areas with low electrification rates. Based on the TNL, we found that the city primacy index of Dar es Salaam was remarkably high and steadily rising. In our research on the power infrastructure of Tanzania's major cities, we found that the land electrification rate (LER) decreased from 2015 to 2019 in most cities, indicating that the power infrastructure gaps are growing in major cities. In addition, we found a negative correlation between the change rate of land electrification and the urban expansion rate, indicating that power infrastructure construction lags behind urbanization. Furthermore, we found a significant decline in the LER in cities that experienced great natural disasters. Further research could examine the spatiotemporal patterns of the LER in a broader range of less-developed regions. Based on this, researchers can identify and monitor cities that are at risk of unexpected disasters, providing support for humanitarian assistance and sustainable regional development.

**Author Contributions:** Conceptualization, C.Z. and X.L.; methodology, C.Z. and Y.R.; supervision, X.L.; writing—original draft, C.Z. and Y.R.; writing—review and editing, X.L. All authors have read and agreed to the published version of the manuscript.

**Funding:** This work was supported by the National Key R&D Program of China under Grant 2019YFE0126800 and the Project of Innovation and Entrepreneurship Training of National Undergraduate of Wuhan University (GeoAI Special Project) under Grant S202110486218.

**Acknowledgments:** The authors would like to thank the anonymous reviewers and handling editors for their constructive comments.

**Conflicts of Interest:** The authors declare no conflict of interest.

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
