# Peer review of "Assessment of Socioeconomic Dynamics and Electrification Progress in Tanzania Using VIIRS Nighttime Light Images"

_remotesensing, doi:10.3390/rs14174240_

Round 1
Reviewer 1 Report
Review of “Assessment of Socioeconomic Dynamics in Tanzania Using 2 VIIRS Nighttime light Images”:
This paper aims to access the socioeconomic dynamics in Tanzania using nighttime light images from the Visible Infrared Imaging Radiometer Suite (VIIRS). Some concerns are as follows:
A1. The abstract part should be rearranged as the tutorial in https://mitcommlab.mit.edu/cee/commkit/abstract/, In the current version, only the general background is provided, without any specific background or knowledge gap. Nighttime light has been thoroughly used for assessing the socioeconomic dynamics, but what is the bottleneck or difficulty in this community, and how you solve it.
A2. The reason behind the finding should be analyzed, why there is a negative correlation between the change rate of land electrification and the urban expansion rate. In common sense, it should be positive correlation. Besides. the conclusion is a little weak, “low-electrification countries like Tanzania nighttime light data can still be applied to access socioeconomic dynamics”, is there any research asserting that nighttime light data is not suitable to assess the socioeconomic dynamics in low-electrification region?
A3. What do you mean by power infrastructure, does it mean common building and street lamp, or does it mean electric station?
A4. According to research conducted in Burkina Faso, most settlements in areas with low electrification rates and low population density are undetectable to nighttime light sensors. So, what do you do to make the low-electrification area detectable?
A5. In fact, the nighttime light is highly related to impervious surface, the light from the buildings and street lamp are commonly adhered to the urban greenbelt. Therefore, in future work, the urban green space dataset (such as DOI: 10.1016/j.jag.2021.102667) may also be used as an auxiliary information for solving the undetectability problem in low-electrification region.
Reviewer 2 Report
The paper deals with an interesting theme highly topical in the context of the current World research trends. The authors focused to representation of accesses the socioeconomic dynamics in Tanzania using nighttime light images. I appreciate especially that authors in this study employed speficic data and applied multiple methods: especially nighttime light data which were collected by the Day/Night Band (DNB) sensors of the Visible Infrared Imaging Radiometer Suite (VIIRS) from 2012 to 2020 and used in this study. Also this study utilized a variety of auxiliary data, including administrative division data, transportation network data, the urban boundary polygons and population data, precipitation data and digital elevation model data (DEM) and GDP data, etc. They used also specific relevant methods or approaches of Regional Economic Disparity and Land Electrification Rate as well as the interpretation of the obtained results. Authors found a negative correlation between the change rate of land electrification 20 and the urban expansion rate, indicating that the power infrastructure popularization is decreased with the increase in urban expansion. They conclude that in low-electrification countries like Tanzania, nighttime light data can still be applied to access socioeconomic dynamics.
The title of the paper is acceptable and adequate and no major changes are necessary. I find the abstract acceptable and well structured. The manuscript has a sufficient scientific value and the information provided represents widening of knowledge. The conclusions are based entirely on the results and the methods used are adequate. The relation between the scientific value and the extent is acceptable. The language and style of the text are at an acceptable level. The tables and illustrations used in the paper are adequate; however I consider the number of references incomplete. The topic dealt with in the paper is also covered by other authors in papers.
I recommend language correction of the text by a native speaker, if possible. I have no other remarks of a rather significant nature concerning the paper. The results are valuable and the scientific paper brings new original data. The manuscript is acceptable after minor revision with minor amendments required; no re-review is necessary. I recommend the paper for the print.
So no elements that should be corrected:
Conclusion: any limitation of your research? So please add it.
I recommend amending the references. This issue is also covered by the newer papers from other authors. I recommend adding some papers into the references.
Figure 10 – text is very small /little poor quality (especially text of legends).
As you see, there is not too much to correct according to my opinion.
Good luck in your future scientific work.
Reviewer 3 Report
With a case study of Tanzania, the manuscript investigates the potential application of VIIRS imagery in areas with low urbanization and weak economic development, based on black marble products. The structure and logic of the manuscript is complete and clear, but it seems to lack sufficient knowledge innovation. In addition, I note that the manuscript sets different distance thresholds in several steps of the analysis, but lacks an adequate and convincing explanation for the setting of these thresholds. Finally, I think the discussion on manuscripts is a bit weak. Please consider refining it.
Reviewer 4 Report
This paper try to assess the socioeconomic Dynamics in Tanzania Using VIIRS Nighttime light Images. Several problems were studied in this paper. Some problems are important. However, the writing of this paper was not good and should imporved before publish.
1) The article discusses too many problems, which makes the goal of the article unclear. It is difficult for readers to understand what the article specifically wants to explain. The author is suggested to dig deeply around a major problem.
2)In several places, for example in the introduction, it was mentioned that low electrification rates and low population density are undetectable to nighttime light sensors. However, in section 4.1.1 the results showed that there is a strong positive correlation between the GDP and the TNL. This is self contradictory. And the author should focus on the problem mentioned in the introduction.
3) The author uses several indicators to analyze economic problems. Why choose these indicators? What is the use of these indicators? What economic problems are these indicators related to?. In particular, are these indicators useful for Tanzania's economic planning and policy-making? The high proportion of the capital economy and regional imbalance are common phenomena, which should not be the focus of Tanzania's economic development at present.
4) the area with positive radiance on nighttime light image is the area with electricity infrastructure. It is not correct. Even in areas without human beings, the night light data has a very small value, such as about 5 in the wilderness.
5) please check the format. Line 510, it is not correct.
6) There is a lack of relevant important literature, especially those published in the last two years.
7) There are a lot discussion in the results section. It shouled moved to the discussion section.
Round 2
Reviewer 1 Report
The authors have addressed the comments from the reviewers.
Author Response
Thank you for constructive comments to improve our manuscript.
Reviewer 3 Report
Thank you for taking my suggestions into consideration and making the appropriate adjustments. Now, it looks much improved. So, I have no additional comments. Congrats!
Author Response

(The authors gave the same response as above.)

Reviewer 4 Report
Some of my previous comments have not been revised. The introduction needs to be rewritten to focus on specific issues.
Author Response
We have rewritten the introduction to focus on specific issues.